# Amino Acid Profiles in Peach (*Prunus persica* L.) Fruit

**DOI:** 10.3390/foods11121718

**Published:** 2022-06-12

**Authors:** Meng Sun, Bintao Zhao, Zhixiang Cai, Juan Yan, Ruijuan Ma, Mingliang Yu

**Affiliations:** 1Institute of Pomology, Jiangsu Academy of Agricultural Sciences (JAAS), Nanjing 210014, China; sm183495665@163.com (M.S.); zbt17730869939@163.com (B.Z.); czxly05@163.com (Z.C.); yanjuanjaas@aliyun.com (J.Y.); marj311@163.com (R.M.); 2Centre for Viticulture and Oenology, Faculty of Agriculture and Life Sciences, Lincoln University, Christchurch 7674, New Zealand; 3College of Horticulture, Nanjing Agricultural University, Nanjing 210095, China

**Keywords:** peach, nectarine, amino acids, sugars, organic acids

## Abstract

Amino acids play an interesting and important role in the metabolism of peaches. The objectives of this study were to investigate and compare amino acid profiles in peaches at harvest for future research about the resistance effects, nutritional value of amino acids in peaches and to produce high-quality peach wine. In the study, 10 peaches and nectarines, including white, yellow and red flesh varieties, were selected for amino acid concentration and composition by high performance liquid chromatography (HPLC). Results showed sugar levels in nectarines were higher than in peaches in this study. High concentrations of total acids were found in “Tropic Prince”, “Yixianhong”, “NJN76” and “Hongrou1”. Malic acids had the highest concentrations, compared toquinic and citric acid concentrations. Total amino acids in yellow and white flesh varieties were over 1100 µg/g FW, while red flesh varieties had total amino acids below 750 µg/g FW. Asn was the highest concentration compared to other amino acids, with the high concentration of Asn in “Tropical Prince’ (3279.15 µg/g FW) and the lowest concentration in “Touxinhong” (559.60 µg/g FW). “Jinxia”, “Yuhua3” and “Chengxiang” had better amino acid scores compared with others, in particularly the lowest value in the red flesh varieties. Finally, according to PCA and the heatmaps, amino acids in “Chengxiang”had evident differences to other varieties, which showed the different amino acid concentrations and composition. Overall, the results of this study highlighted three yellow flesh and one white flesh varieties that had satisfactory concentrations and components of amino acid values. In addition, amino acids were the precursors of aroma compounds, so these differences between varieties werea new way to screen the potential varieties for producing high quality peach wines with the anticipated specific characteristics.

## 1. Introduction

The peach originated in China and is widely planted over 100,000 hectares. *Prunuspersica* L. Batsch (peach) is the most important fruit in China [1]. Amino acids are part of the primary constituents of peaches. Amino acids and their biosynthesis are also important for all living things.They are the subunits for proteins, enzymes and nucleic acids, and play roles in anabolism as signalling molecules and osmotic adjustment, such as leaf stomatal opening, ion transport regulation and redox homeostasis [2,3].

Previous studies about peaches mainly focused on accumulation of sugar, acids and phenolic and aroma compounds [4,5]. The total amino acid concentration in peaches increases during the fruit’s development [6]. Amino acids serving as the precursors of proteins could potentially play roles as plant defence in response to stresses [7,8]. Essential amino acids are one of the most important nutrients to evaluate the protein quality of peaches and nectarines [5].

However, there was less research about amino acid profiles in peaches to screen the potential varieties for producing high quality peach wines with the anticipated specific characteristics. Amino acids are energy sources for yeast and bacterial metabolism [2]. In addition, during fermentation, amino acids are precursors of aromatic compounds being metabolised into higher alcohols, aldehydes, organic acids, phenols and lactones [9]. Previous research found the total phenolic compound contents, total flavonoid contents, and antioxidant activity in “Redhaven” peach wine were higher than those of white wines [10]. Peach wine seems to be more promising and popular with consumers due to its delicate flavour and nutritive value. Moreover, the sensory evaluation results showed that peach wine was more acceptable to consumers than grape white wines and had good marketing prospects [10]. The organoleptic evaluation of the prepared yellow peach wine showed that the prepared yellow peach wine retained the clear aroma of the peach fruit, mellow and estery aromas.The increases in esters and alcohols during the fermentation process enhanced a strong alcoholic flavour in the yellow peach wine [11].

Thus, the objectives of this study were to investigate and compare amino acid profiles in peaches at harvest for future research about the resistance effect and nutritional value of amino acids in peach trees and to produce high-quality peach wine with the anticipated specific characteristics.

## 2. Materials and Methods

### 2.1. Plant Material

The early-ripening peach and nectarine varieties were selected as below (Table 1). The undamaged and healthy nectarine, peach and flat fruit were harvested from the Peach Experimental Orchard of Nanjing (32°2′ N, 118°52′ E), when crops stopped enlarging and began softening at commercial ripening in June 2021.

### 2.2. Sample Collection

Five fruits from each tree (5 fruits per tree × 2 trees) were collected from different sites (top, medium and bottom) of trees at harvest. A sample collection of 10 fruits was divided into three blocks and 5 g of flesh was collected from each block. Samples were stored in −80 °C frozen for the analysis of amino acids.

### 2.3. Amino Acids Analysis

The flesh was ground with 600 μLof 10% formic acid methanol and two steel balls in grinds for 90 s at 60 Hz and centrifuged for 5 min at 12,000 rpm and 4 °C. The 20 μL supernatant was diluted with 380 μL of 10% formic acid methanol in a new tube. The samples were filtered through a 0.22 μm nylon syringe filter into an HPLC glass vial and capped tightly. An internal standard, γ-aminobutyric acid (γ-GABA), was added to a final concentration of 100 μmol/L. For inline-derivatisation of the primary amino acids, ơ-phthaldialdehyde was used as a fluorescence derivative; iodoacetic acid/mercaptopropionic acid was used to increase cysteine sensitivity; and 9-fluorenylmethyl chloroformate was a fluorescence derivative for proline.

The method of chromatography was followed and modified by Gregan et al. [12]. The samples were injected into an HPLC system with a 100 × 2.1 mm, 1.7 μm C18 column (Waters ACQUITY UPIC, Milford, MA, USA) thermostatized at 16 °C. Data were analysed using the Chemstation (Agilent, Santa Clara, CA, USA) chromatography data system. The mobile phase consisted of two solvents: solvent A (10% methanol with 0.1% formic acid) and solvent B (50% methanol with 0.1% formic acid). The gradient programme was 0–6.5 min, 10–30% B; 6.5–7 min, 30–100% B; 7–8 min, 100% B; 8–8.5 min, 100–10% B; 8.5–12.5 min, 10% B, with a flow rate of 0–8.5 min, 0.3 mL/min; 8.5–12.5 min, 0.3–0.4 mL/min. For detection, a fluorescence detector with an excitation at 335 nm and emission at 440 nm. At 7 min, the detector was switched to a second channel (excitation at 260 nm and emission at 315 nm) to detect proline. Amino acids were identified by their retention times and their concentrations were calculated in parallel to calibrate the internal amino acid standard (γ-GABA, 100 μmol/L).

### 2.4. Flesh Parameters

Using 5 mL of extracting solution (ethanol: 0.2% metaphoric acid *v/v*), 0.5 g of the flesh was ground and extracted by ultrasound wave for 1 h. The solution was centrifuged for 5 min at 10,000 rpm and 4 °C. The 0.8 mL supernatant was dried by a concentrator (Eppendorf, Concentrator plus, Sigma Aldrich, Saint Louis, MO, USA, 230 V/50–60 Hz) for 2 h. 1.6 mL of ultrapure water was added into the concentrate to obtain sugar/organic acid solution.

The method of chromatography was followed and modified byShen et al. [13]. The samples were filtered through a 0.22 μm nylon syringe filter into an HPLC glass vial and capped tightly. Then, 0.5 μL of the samples were injected into an HPLC system with a 10 μm, 250 × 6 mm CARBOSep CHO-620 CA column (Transgenomic, Omaha, NE, USA) for soluble sugars at 80 °C with a differential refractive index detector. The mobile phase used ultrapure water for 0.5 mL min^−1^. Ultrapure water was used as a standard for each batch of samples. Ultrapure water (10, 5, 1, 0.5, 0.1 g/L) was used to establish a standard curve for reporting soluble sugars.

Next, 0.5 μL of the samples were injected into an HPLC system with a 5 μm, 250 × 4.6 mm ZORBAX Eclipse XDB-C18 column (Agilent, Santa Clara, CA, USA) for organic acids at 2 °Cwith a VWD UV detector (λ = 214 nm). The mobile phase was 0.02 mol/L KH_2_PO_4_ (pH 2.7) for 0.5 mL min^−1^. Data were analysed using the Chemstation (Agilent) chromatography data system. KH_2_PO_4_ was used as a standard for each batch of samples. 0.02 mol/L KH_2_PO_4_ (pH 2.7) (3.333, 1.667, 0.333, 0.167, 0.033 g/L)was used to establish a standard curve for reporting organic acids.

Soluble sugars and organic acids were identified by their retention time and their concentrations were calculated in parallel to calibrate the internal ultrapure water and 0.02 mol/L KH_2_PO_4_ (pH 2.7) standard curves.

### 2.5. Protein Quality Calculations

The amino acid score was calculated by comparing the individual amino acids of each test article with the recommended Food and Agriculture Organization of the United Nations (FAO)/World Health Organization (WHO) reference pattern, mg/g protein (FAO, 2013). Both amino acids and reference patterns were first expressed in mg amino acids/g protein units. The lowest calculated amino acid ratio (limiting amino acids) was considered as the amino acid score [14].

### 2.6. Statistics

Statistical analysis was undertaken using R studio (v3.3.2). Amino acid data for samples from 10 varieties were analysed using thepheatmap package for agglomerate hierarchical clustering and the ropls package for PCA. The correlation between amino acids and sugars/acids was conducted by thecor.test of R studio (v3.1.3) (TUNA team, Beijing, Tsinghua University).

## 3. Results

At harvest, carbon compounds, acids and amino acids were measured in peaches or nectarines collected at harvest in June 2021 (Table 2 and Table 3).

### 3.1. Flesh Parameters

Different varieties presentedsignificant differences in the concentrations of carbon compounds (sucrose, glucose, fructose and sorbitol) and acids (quinic, malic and citric acids) (Table 2). In general, this study showed that nectarines (1.“Jin Xia” and 7.“NJN76”) had higher sugar levels than peaches. Sucrose in “Jinxia” and “NJN76” were higher than others, 43.98 and 40.63 mg/g, respectively. ‘Tropic Prince”, “Dahezaosheng” and “Chengxiang” had lower sucrose concentrations than others, between 20 and 30 mg/g. Glucose concentration ranged from 7.59 to 14.13 mg/g with the lowest concentration in “Chengxiang”, and the highest concentration in “Yixianhong”. However, “Chengxiang” had the lowest fructose concentration (10.43 mg/g), whereas the highest fructose concentration (17.76 mg/g) was in “Touxinghong”. Sorbitol concentrations in all varieties were the lowest than other sugars, below 4.24 mg/g. In 10 varieties, high concentrations of total acids were “Tropic Prince” (10.43 mg/g), “Yixianhong” (13.42 mg/g), “NJN76” (10.45 mg/g) and “Hongrou1” (10.40 mg/g). Malic acids had the highest concentrations compared toquinic and citric acid concentrations.

### 3.2. Amino Acids

At harvest, our study showed total amino acid concentrations of 6 varieties from 1 to 6 were over 1300 μg/g FW, while two red flesh varieties (“Hourou1” and “Touxinhong”) were 723.22 and 662.70 μg/g FW, respectively. In addition, two yellow flesh varieties 

“Jinxia” and “Tropic Prince”, and one white flesh variety “Yixianhong”, had higher total amino acid concentrations (over 2800 μg/g FW) than other varieties at harvest. In *α*-ketoglutarate family, concentrations of Pro, Glu and Gln in peaches were higher than Arg and His. Our results showed the magnitudes of Glu and Gln levels were different between varieties. There was 145.03 µg/g FW of Glu and 627.32 µg/g FW of Gln in “Yuhua 3”, compared to 24.61 µg/g FW of Glu and 6.72 µg/g FW of Gln in “Touxinhong”. “Tropical Prince” and “Chengxiang” also had high Gln concentrations, 111.91 and 106.03 µg/g FW, respectively. Thus, Arg levels were very low, below 3 µg/g FW. In these varieties, high concentrations of Pro were “Jinxia”, “Yuhua 3” and “Tropic Prince”, which were reached at 65.26, 52.02 and 41.72 µg/g FW, while other varieties had Pro concentrations below 15 µg/g FW. In the shikimate family, these amino acids were precursors of aroma compounds in fruits. Phe, Trp and Tyr concentrations, however, were low in 10 varieties. “Jinxia” (14.64 µg/g FW), “Yuhua 3” (16.22 µg/g FW), “Tropic Prince” (16.72 µg/g FW) and “Chengxiang” (17.58 µg/g FW) had higher concentrations of shikimate family than other six varieties. GABA concentrations reached at 20.16 and 14.04 µg/g FW in “Jinxia” and “Shazizaosheng”, respectively. In the aspartate family, Asnhad the highest concentration of all the amino acids, and most varieties had over 900 µg/g FW of Asn, except “Hongrou1” and “Touxinhong”. “Tropical Prince” had the high concentration of Asn (3279.15 µg/g FW), while two red flesh fruits, “Hongrou1” and “Touxinhong”, had low concentrations of Asn (611.16 and 559.60 µg/g FW).

Amino acids build up proteins in peach. They can also be divided into essential (Arg, His, Phe, Leu, Val, Thr, Ile, Met and Lys) and non-essential amino acids (Pro, Glu, Tyr, Ala, Asp, Ser and Gly). It was well-known that there were not enough proteins and essential amino acids in most fruits. The results were comparable to the ones documented by Botoran et al. [15], concerning the amino acid profile of peaches. The amino acid score method was used to evaluate the protein quality of peaches or nectarines in this study. Most essential amino acids were quite limited, which cannot contribute to synthesis the necessary quality proteins for adults (Table 4). “Jinxia”, “Yuhua3” and “Chengxiang” have better amino acid scores compared with others, in particular the lowest value in the red flesh varieties. Total amino acid concentrations were low in red varieties, while yellow flesh varieties had high concentrations.

### 3.3. PCA and Correlation

The score plot of the first two principal components (PC1 and PC2) (Figure 1). The score plot was a map of 10 varieties of peaches and nectarines. The distribution points close to each other had similar amino acid profiles, whereas those far from each other were dissimilar. The results showed that the two principal components explained 73.0% of the total variance. The first principal component (PC1) explained 56.4% of the total variance, while PC2 explained 16.6%. The parameters chosen for PCA included 1. “Jinxia”, 2. “Yuhua 3”, 3. “Tropic Prince”, 4. “Dahezaiosheng”, 5. “Yixianhong”, 6. “Chengxiang”, 7. “NJN76”, 8. “Shazizaosheng”, 9. “Hongrou1” and 10. “Touxinhong”. 1, 2, 3 and 5 were located together in the lower right-hand corner, thus representing one group of varieties with some similarity in amino acid profiles. In addition, 1, 4, 5, 7 and 8 were close to the center (origin) of the plot, which indicated they had average properties. Moreover, 9 and 10 were very close to each other in the down left-hand corner, so they had similar amino acid profiles. They were red flesh varieties. Parameter 6 was far away from other points and the centre (origin) of the plot, which showed the different amino acid concentrations and composition.

According to this heatmap (Figure 2), 6 had an evident difference to other varieties. Although 1, 3 and 6 were yellow flesh varieties, 6 had a strong fruity smell compared to 1 and 3. Heat maps simultaneously visualise clusters of samples andfeatures. First, hierarchical clustering was done of the rows of amino acids and the columns of varieties matrix. Then the branches of the dendrograms were rotated so that the blocks of “high” and “low” concentration values were adjacent in the amino acid matrix. The branches of the trees were rotated to create blocks in which the individual values were close in both directions. Finally, a colour scheme was applied for the visualisation and the amino acid matrix was displayed.These were colour-coded by amino acid concentrations. The heat map used red and purple for colours for this study. Red was for high concentrations of amino acids, while purple was for low amino acid concentrations. A number of amino acids were high in 1, 2, 3, 5 and 6,and pretty low in other varieties, particularly 9 and 10 (two red flesh peaches).

The correlation of free amino acids, sugar and acid concentrations was presented in Figure 3. It can be seen that 1, 0 and −1 showed positive correlation (red), no correlation (white) and negative correlation (blue), respectively. Amino acids had negative correlation with sugars and acids. The significant negative correlation presented between Ala, Gln, Tyr and sugars.

## 4. Discussion

Sorbitol concentrations were lower than others in peaches, because sorbitol was the main product of photosynthesis in mature peach leaves. In addition, sorbitol changed less and remained relatively low throughout fruit growth. Early spring growth in peach trees was mostly dependent on stored carbohydrates that were mobilized after the release from dormancy and transported to the growing sinks through the xylem (as hexoses) [4]. Glucose and fructose concentrations were very close and lower than sucrose in all varieties. Sucrose was the dominant sugar even in the early stages of fruit growth. During fruit development, glucose uptake was through fruit mesocarp tissue. Sorbitol appeared to be converted to glucose by sorbitol oxidase to support fruit growth. Glucose and fructose represent a transitory storage/metabolic pool [4,16]. High acid levels may provide the acid environment for anthocyanin and carotenoid stable and coloration in the flesh [17]. Peaches and nectarines are malic acid type fruits. Additionally, in peach flesh, there was a net dissimilation of citric acid, while there was a synthesis of malic acids during ripening [18]. In peaches, the double-sigmoidal growth pattern corresponded to two stages of sink activity [19].

Nitrogenous compounds were transferred from senescing leaves to the flesh and pits through xylem and phloem during ripening, and then were converted to amino acids in fruits [8,20]. In this study, the total amino acid concentrations accumulated from 662.70 to 3901.58 μg/g FW, which were not consistent with previous study ranging from 1000 to 1300 μg/g FW in the peach flesh at harvest [21].Moreover, the concentrations and components of individual amino acids in peach flesh were immensely different to varieties in the study. The inconsistent results may reflect their different genetic backgrounds [22]. Asn accounted for the bulk of amino acid content (over 50% of the non-protein and protein nitrogen) in the peach flesh during development. This was in agreement with previous studies [19]. The accumulation of Asn could attribute to the increase in synthesis during the early fruit development and the decrease in catabolism of Asn in ripening [19,23]. Asn biosynthesis was induced by the amount of *β*-Cyano-Alanine hydratase and *β*-Cyano-Alanine synthase. The decreases in two asparaginases (M5WUV5 and M5X0K4) reduced the catabolism of Asn, resulting in the accumulation of amino acids. Asn accumulation in the early development and ripening provides [23].

Not only Asn but also Gln, as a large proportion of the nitrogenous compounds, was imported into fruits [24]. Pro biosynthetic pathways were from Glu via many enzymes. Gln was converted into other amino acids by aminotransferases, such as Pro and Arg, which occupied the greatest percentages of total amino acids in peach. During dormancy, Arg was the largest percentage of these reserve soluble nitrogenous compounds in the roots and crown. Then, it was mobilised and transferred to the xylem with water in the transpiration stream in spring [19,25]. Pro can be used as an index in response to environmental stresses. Its concentrations reflected the different resistance of varieties under the same environmental conditions [26]. During fermentation, yeast utilized a great quantity of amino acids. Arg, Ala, Asp, Glu, Thr and Ser were preferred nitrogen sources for wine yeasts [27].

Shikimate family was the precursors of volatile compounds. “Jinxia”, “Yuhua3”, “Tropic Prince” and “Chengxiang” had higher levels of shikimate family than others, in consistency with the previous study [28]. Thus, yellow flesh fruit may have a more pleasantodour than others after fermentation, resulting from more precursors of volatile compounds [29]. “Tropic Prince” and “Chengxiang” had higher concentrations the branched amino acids (Val and Leu), which could be used by alcoholic fermentation [30]. These high concentrations of GABA reflected low concentrations of citric acid and total acids. This was due to up-regulation of a GABA pathway gene to stimulate the citrate degradation [31].

According to the protein quality evaluation of peach, this study showed that the yellow flesh fruit had a higher value than white and red flesh fruit. Thus, yellow flesh varieties contributed to providing the necessary amounts of quality protein for adults and provided real nutritional quality. In PCA, “Chengxiang” was far away from the centre, resulting from the different components of amino acids. In addition, it may reflect different amino acids as precursors for the biosynthesis of volatile compounds in it [32]. The significant negative correlation was between Gln and glucose and fructose. Amino acids can act as precursors or intermediates for organic acids in the Krebs cycle during late development. Glucose provides the carbon skeletons into the Calvin cycle for amino acid accumulation [6,33].

## 5. Conclusions

This study highlighted the differences in concentration and components of amino acids in 10 peach and nectarine varieties. In this aspect, valuable information was that sugar levels in nectarines (“Jinxia” and “NJN76”) were higher than in peaches. Glucose and fructose concentrations were very close and lower than sucrose in all varieties, so sucrose was the dominant sugar in 10 varieties. High concentrations of total acids were found in“Tropic Prince”, “Yixianhong”, “NJN76” and “Hongrou1”. Malic acids had the highest concentrations compared toquinic and citric acid concentrations. Ans and Glu account for the bulk of total amino acids, and red flesh varieties had lower total or individual amino acid concentrations than yellow and white flesh varieties. “Jinxia”, “Yuhua3” and “Chengxiang” had better amino acid scores compared with others, in particular the lowest value in the red flesh varieties. Finally, according to PCA and the heatmaps, amino acids in “Chengxiang”had an evident difference to other varieties, which showed the different amino acid concentrations and composition. Therefore, this study was critical to understanding amino acid profiles of metabolites and transcripts and provide novel insights into amino acids of resistance research and nutrition aspects in peaches. Furthermore, amino acids could be precursors of aromatic compounds, so this study may evaluate the potential varieties for producing high quality peach wines with the anticipated specific characteristics.

## Figures and Tables

**Figure 1 foods-11-01718-f001:**
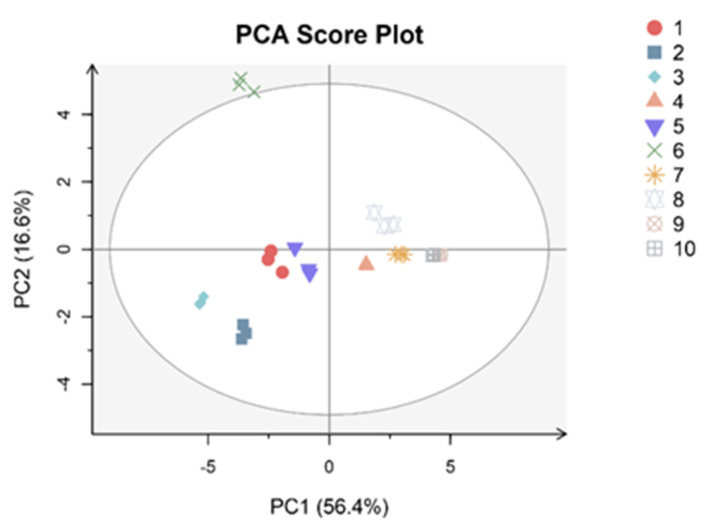
Principal component analysis (PCA) of amino acid data of 10 peach and nectarine varieties in a 2D graph of PC1 and PC2. 1 indicates Jinxia; 2 indicates Yuhua3; 3 indicates Tropic Prince; 4 indicates Dahezaosheng; 5 indicates Yixianhong; 6 indicates Chengxiang; 7 indicates NJN76; 8 indicates Sazizaosheng; 9 indicates Hongrou1; 10 indicates Touxinhong.

**Figure 2 foods-11-01718-f002:**
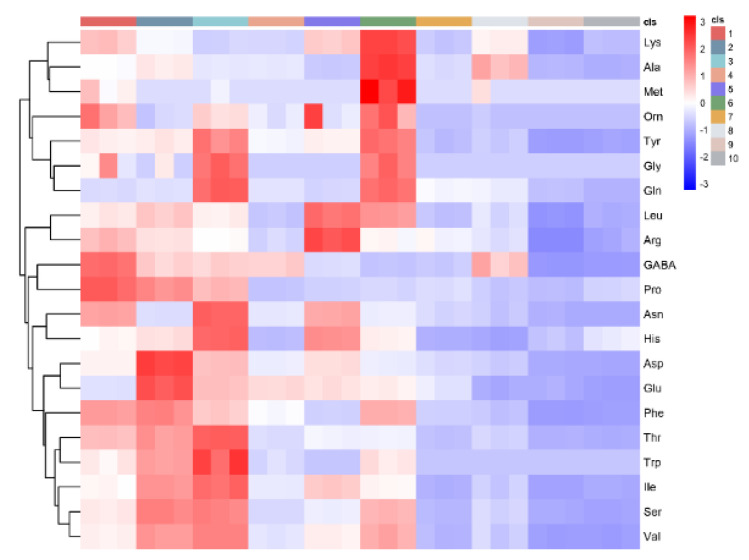
Heat-map of amino acid concentration values in 10 peach and nectarine varieties. 1 indicates Jinxia; 2 indicates Yuhua3; 3 indicates Tropic Prince; 4 indicates Dahezaosheng; 5 indicates Yixianhong; 6 indicates Chengxiang; 7 indicates NJN76; 8 indicates Sazizaosheng; 9 indicates Hongrou1; 10 indicates Touxinhong.

**Figure 3 foods-11-01718-f003:**
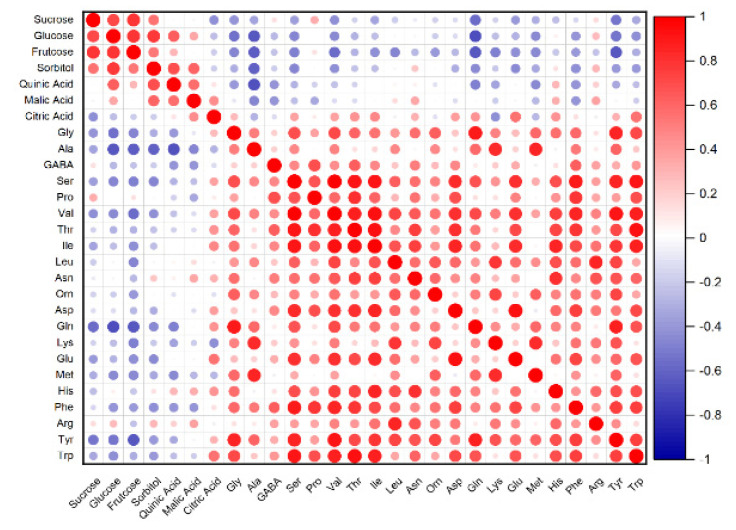
Correlation matrix heat maps. Heat map of Spearman correlation between amino acids and sugars and organic acids between each biomarker for the overall population analyzed.

**Table 1 foods-11-01718-t001:** Peach (*Prunus persica* L.) and nectarine varieties used in this study.

	Variety	Type	Flesh Colour
1.	Jinxia	nectarine	yellow
2.	Yuhua 3	peach	white
3.	Tropic Prince	peach	yellow
4.	Dahezaosheng	peach	white
5.	Yixianhong	peach	white
6.	Chengxiang	peach	yellow
7.	NJN76	nectarine	white
8.	Sazizaosheng	peach	white
9.	Hongrou1	peach	red
10.	Touxinhong	peach	red

**Table 2 foods-11-01718-t002:** Sugar and acid levels of 10 peach and nectarine varieties at harvest (mg/g).

Varieties	Sucrose	Glucose	Fructose	Sorbitol	Quinic Acid	Malic Acid	Citric Acid	Total Acids
1. Jinxia	43.98 ^a^	12.06 ^b^	15.27 ^e^	3.70 ^b^	1.90 ^cd^	2.43 ^e^	0.58 ^g^	4.91 ^f^
2. Yuhua3	32.67 ^c^	10.47 ^d^	13.57 ^cd^	1.10 ^f^	1.57 ^bc^	2.28 ^f^	2.11 ^c^	5.96 ^de^
3. Tropic Prince	25.67 ^e^	8.50 ^e^	12.09 ^b^	2.06 ^d^	1.98 ^cd^	5.15 ^c^	3.30 ^a^	10.43 ^b^
4. Dahezaosheng	23.82 ^f^	8.52 ^e^	11.65 ^ab^	1.27 ^f^	2.04 ^cd^	3.58 ^d^	2.11 ^c^	7.72 ^c^
5. Yixianhong	32.91 ^c^	14.13 ^a^	12.97 ^bc^	4.24 ^a^	4.23 ^a^	7.74 ^a^	1.45 ^e^	13.42 ^a^
6. Chengxiang	23.63 ^f^	7.59 ^f^	10.43 ^a^	1.12 ^f^	1.09 ^e^	2.82 ^e^	1.28 ^ef^	5.20 ^ef^
7. NJN76	40.63 ^b^	11.54 ^b^	14.87 ^de^	2.90 ^c^	1.63 ^d^	6.46 ^b^	2.36 ^b^	10.45 ^b^
8. Shazizaosheng	34.46 ^c^	9.22 ^e^	12.54 ^bc^	1.76 ^e^	1.07 ^e^	1.78 ^f^	0.38 ^h^	3.22 ^g^
9. Hongrou1	30.10 ^d^	11.02 ^c^	14.51 ^de^	3.75 ^b^	3.82 ^a^	4.88 ^c^	1.71 ^d^	10.40 ^b^
10. Touxinhong	39.15 ^b^	14.07 ^a^	17.76 ^f^	2.70 ^c^	2.62 ^b^	2.76 ^e^	1.15 ^f^	6.53 ^d^
*p* value	<0.001	<0.001	<0.001	<0.001	<0.001	<0.001	<0.001	<0.001

The data are presented as the mean (*n* = 3). Different letters within each column indicate significant differences between varieties according to Tukey’s test (*p* < 0.05).

**Table 3 foods-11-01718-t003:** Amino acids of 10 peach and nectarine varieties at harvest (µg/g).

Amino Acid (µg/g)	*α*-Ketoglutarate					Shikimate (Aromatic)			Pyruvate			Aspartate						3-phosphoglycerate		Others		
	Pro	Arg	Glu	Gln	His	Phe	Trp	Tyr	Leu	Val	Ala	Asp	Asn	Thr	Ile	Met	Lys	Ser	Gly	Orn	GABA	Total
1. Jinxia	65.26 ^a^	1.60 ^b^	54.17 ^e^	24.62 ^de^	4.23 ^d^	10.15 ^b^	0.49 ^c^	4.00 ^b^	3.20 ^d^	13.72 ^d^	20.19 ^d^	108.78 ^d^	2566.68 ^b^	28.08 ^c^	5.65 ^d^	0.21	5.52 ^b^	45.78 ^c^	1.70 ^b^	0.47 ^ab^	20.16 ^a^	2984.66 ^b^
2. Yuhua3	52.02 ^b^	1.26 ^c^	145.03 ^a^	27.32 ^cde^	5.16 ^c^	11.12 ^a^	0.98 ^b^	4.12 ^b^	3.93 ^c^	22.76 ^b^	24.74 ^c^	240.56 ^a^	1126.78 ^de^	33.20 ^b^	10.85 ^b^	N.A	3.28 ^d^	77.13 ^a^	N.A	0.08 ^c^	12.18 ^b^	1802.50 ^c^
3. Tropic Prince	41.72 ^c^	1.02 ^d^	97.07 ^b^	111.91 ^a^	9.26 ^a^	7.86 ^d^	1.52 ^a^	7.34 ^a^	3.04 ^d^	25.59 ^a^	15.09 ^e^	148.13 ^b^	3279.15 ^a^	43.21 ^a^	13.27 ^a^	N.A	2.08 ^e^	77.60 ^a^	4.04 ^a^	0.29 ^ab^	12.42 ^b^	3901.58 ^a^
4. Dahezaosheng	7.89 ^fg^	0.65 ^e^	85.17 ^c^	29.05 ^cd^	1.75 ^f^	5.06 ^e^	0.13 ^d^	3.09 ^c^	1.26 ^f^	9.24 ^e^	15.18 ^e^	84.76 ^e^	1234.31 ^cd^	12.52 ^e^	4.11 ^e^	N.A	2.18 ^e^	26.46 ^e^	N.A	0.11 ^bc^	12.52 ^b^	1535.43 ^d^
5. Yixianhong	10.97 ^def^	2.54 ^a^	82.48 ^cd^	22.62 ^e^	7.78 ^b^	3.02 ^f^	N.A	3.86 ^b^	6.08 ^a^	13.84 ^d^	8.50 ^f^	124.93 ^c^	2517.30 ^b^	16.07 ^d^	8.18 ^c^	N.A	5.27 ^b^	33.24 ^d^	N.A	0.31 ^ab^	6.21 ^c^	2873.20 ^b^
6. Chengxiang	12.46 ^d^	1.04 ^d^	76.12 ^d^	106.03 ^a^	4.52 ^d^	9.26 ^c^	0.55 ^c^	7.77 ^a^	5.14 ^b^	21.13 ^c^	63.14 ^a^	83.98 ^e^	1318.34 ^c^	16.05 ^d^	5.75 ^d^	0.90	9.56 ^a^	61.89 ^b^	3.79 ^a^	0.54 ^a^	4.39 ^c^	1812.34 ^c^
7. NJN76	14.22 ^d^	0.96 ^d^	56.41 ^e^	37.48 ^b^	1.29 ^g^	2.91 ^f^	N.A	1.37 ^d^	1.16 ^f^	4.37 ^g^	12.69 ^e^	69.99 ^f^	1061.27 ^e^	8.21 ^f^	1.03 ^g^	N.A	1.72 ^ef^	16.34 ^f^	N.A	N.A	4.74 ^c^	1296.18 ^e^
8. Shazizaosheng	8.69 ^efg^	0.74 ^e^	30.06 ^f^	33.33 ^bc^	0.91 ^g^	2.44 ^f^	N.A	1.81 ^d^	1.84 ^e^	7.14 ^f^	36.95 ^b^	60.08 ^g^	907.75 ^f^	11.35 ^e^	2.63 ^f^	N.A	4.08 ^c^	25.98 ^e^	N.A	N.A	14.04 ^b^	1149.83 ^f^
9. Hongrou1	5.64 ^g^	N.A	30.97 ^f^	13.15 ^f^	2.02 ^f^	0.42 ^g^	N.A	0.34 ^e^	0.09 ^h^	1.64 ^h^	4.59 ^g^	33.26 ^h^	611.16 ^g^	5.80 ^g^	0.41 ^g^	N.A	0.36 ^g^	12.53 ^g^	N.A	N.A	0.85 ^d^	723.22 ^g^
10. Touxinhong	12.10 ^de^	0.26 ^f^	24.61 ^f^	6.72 ^g^	3.32 ^e^	0.60 ^g^	N.A	0.51 ^e^	0.68 ^g^	1.57 ^h^	2.73 ^g^	30.95 ^h^	559.60 ^g^	5.29 ^g^	0.72 ^g^	N.A	1.37 ^f^	10.58 ^g^	N.A	N.A	1.10 ^d^	662.70 ^g^
*p* value	<0.001	<0.001	<0.001	<0.001	<0.001	<0.001	<0.001	<0.001	<0.001	<0.001	<0.001	<0.001	<0.001	<0.001	<0.001	<0.001	<0.001	<0.001	<0.001	<0.001	<0.001	<0.001

The data are presented as the mean (*n* = 3). Different letters within each column indicate significant differences between varieties according to Tukey’s test (*p* < 0.05). N.A, not available.

**Table 4 foods-11-01718-t004:** Protein quality evaluation of peach (*Prunus persica* L.) varieties by amino acid score method (mg/g).

Variety	1. Jinxia	2. Yuhua3	3. Tropic Prince	4. Dahezaosheng	5. Yixianhong	6. Chengxiang	7. NJN76	8. Shazizaosheng	9. Hongrou1	10. Touxinhong
Val	0.35	0.58	0.66	0.24	0.35	0.54	0.11	0.18	0.04	0.04
Met	0.04	0	0	0	0	0.15	0	0	0	0
Lys	0.12	0.07	0.05	0.05	0.12	0.21	0.04	0.09	0.01	0.03
Ile	0.19	0.36	0.44	0.14	0.27	0.19	0.03	0.09	0.01	0.02
Leu	0.05	0.07	0.05	0.02	0.10	0.09	0.02	0.03	0.00	0.01
Phe + Tyr	0.64	0.69	0.69	0.37	0.31	**0.77**	0.19	0.19	0.03	0.05
Thr	**1.22**	**1.44**	**1.88**	0.54	**0.70**	**0.70**	0.36	0.49	0.25	0.23
His	0.28	0.34	0.62	0.12	0.52	0.30	0.09	0.06	0.13	0.22

## Data Availability

All data analysed or generated during this study are available within the manuscript and can be requested from the corresponding author.

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
