# Peer review of "Amino Acid Profiles in Peach (Prunus persica L.) Fruit"

_foods, 2022, doi:10.3390/foods11121718_

Round 1

Reviewer 1 Report

This study is aimed to investigate and compare amino acid profiles and some other parameters in peaches at harvest. The study contains some interesting results. However, the study requires thoughout revisions in several aspects. There are many mistakes in the study and the study is not carefully written.

 L2: Prunus presica – in italic

L12: Amino …

Abstract: Need info from M and M. Descrpition of results is very weak.

L21-24: These two sentence is too general give a mlore concrete conlusion.

L25: Title is missing.

 L68: Table 1: Peach and nectarine varieties used in this study.

Tables 2, 3: Give letter for each value in oder to indciate significant differences among each value. Give also separating value for e.g LSD0.05 value for each parameters. Give the name of the cultivars.

 Table 4: give explanation for 1-10 in the first row.

 Figure 1: Give explanation for 1-10.

 Figures 2 and 3: These figures are too small. Unreadable.

 L292: Conclusion is weak. Give much more suitable conclusions from your gained results.

References: This section need to reformat fully for journal requirements. Scientific name should be in italic.

Author Response

Dear Reviewer,

Thanks for your comments. I have modifies each ponits. 

Kind regards, maya

Reviewer 2 Report

The article presented for review: ‘Amino Acid Profiles in Peach (Prunus persica L.) Fruit is interesting However, some issues need to be clarified or supplemented. The comments are included below.

1. Introduction

- Line 46-48: Previous research found the total phenolic compound contents, total flavonoid contents, and antioxidant activity in ‘Redhaven’ peach wine were higher than those of white wines [10]. - If the authors treat peach wines as yellow wines, please explain which white wines were tested?

- Line 57-59: Thus, amino acids in some potential peach and nectarine varieties may influence in producing high quality peach wines with the anticipated specific characteristics. - The sentence is not consistent with the above information. Needs to be redrafted. The text should form a coherent whole.

Materials and Methods

- Line 87: Was the chromatographic column thermostated, and if so, at what temperature?

Author Response

(The authors gave the same response as above.)

Round 2

Reviewer 1 Report

The authors made acceptable revisions. The study is improved.